# Laser-induced plasmonic colours on metals

Jean-Michel Guay[1,2], Antonino Calà Lesina[1,2], Guillaume Côté[1,2], Martin Charron[1], Daniel Poitras[3], Lora Ramunno[1,2], Pierre Berini[1,2,4] & Arnaud Weck[1,2,5]

Plasmonic resonances in metallic nanoparticles have been used since antiquity to colour glasses. The use of metal nanostructures for surface colourization has attracted considerable interest following recent developments in plasmonics. However, current top-down colourization methods are not ideally suited to large-scale industrial applications. Here we use a bottom-up approach where picosecond laser pulses can produce a full palette of non-iridescent colours on silver, gold, copper and aluminium. We demonstrate the process on silver coins weighing up to 5 kg and bearing large topographic variations ($\sim$1.5 cm). We find that colours are related to a single parameter, the total accumulated fluence, making the process suitable for high-throughput industrial applications. Statistical image analyses of laser-irradiated surfaces reveal various nanoparticle size distributions. Large-scale finite-difference time-domain computations based on these nanoparticle distributions reproduce trends seen in reflectance measurements, and demonstrate the key role of plasmonic resonances in colour formation.

[1] Department of Physics, University of Ottawa, Ottawa, Ontario, Canada K1N 6N5. [2] Centre for Research in Photonics, University of Ottawa, Ottawa, Ontario, Canada K1N 6N5. [3] Information and Communication Technologies Portfolio, National Research Council of Canada, 1200 Montreal Rd. Building M-50, Ottawa, Ontario, Canada K1A 0R6. [4] School of Electrical Engineering and Computer Science, University of Ottawa, Ottawa, Ontario, Canada K1N 6N5. [5] Department of Mechanical Engineering, University of Ottawa, Ottawa, Ontario, Canada K1N 6N5. Correspondence and requests for materials should be addressed to J.-M.G. (email: jguay036@uottawa.ca) or to A.C.L. (email: antonino.calalesina@uottawa.ca) or to A.W. (email: aweck@uottawa.ca).

Metal nanoparticles (NPs) are used in a multitude of applications, but perhaps the oldest is as a colourizing agent when dispersed in a host dielectric, as in the (dichroic) Lycurgus cup[1,2]. Exposed to optical radiation, metal NPs exhibit scattering properties due to excited plasmons that depend on their shape, size, composition and the host medium[3–6]. Producing colours by exploiting plasmonic effects on metal nanostructures[7] is of interest because the colours can last a long time (for example, the Lycurgus cup), can be rendered down to the diffraction limit[8] and can be used in any metal colouring or marking application where inks, paints or pigments should be avoided for environmental, health cost or other reasons.

Fabrication techniques to render plasmonic colours include laser interference lithography[9], electron beam lithography[8,10,11], ion beam lithography or milling[12] and hot embossing or nanoimprint lithography[9,12]. Kumar et al.[8] fabricated coloured images via electron beam lithography having a resolution as high as ≈100,000 dots per inch. However, the production of large coloured surfaces with such techniques is challenging, time consuming and the methods may be incompatible with the demands of low-cost mass manufacturing. Furthermore, such processes generally require flat surfaces, and with the exception of periodic structures below the diffraction limit of visible light[13,14], the colours produced are often angle dependent (iridescent).

Femtosecond (fs) lasers have been used for metal colourization because of their ablation characteristics, for example, the tendency to preferentially release NPs, compared with the large clusters and chunks produced by nanosecond pulses[15–17]. Vorobyev and Guo[18] showed that exposing metals to fs pulses could produce highly absorptive surfaces (the so-called 'black metals') and surfaces producing a specific colour. However, the colour palette that can be produced by such lasers appears to be limited and angle dependent due to the underlying regular structure[19], and the low pulse energy of fs lasers makes the colouring process time consuming.

Picosecond (ps) lasers have lower costs and higher pulse energies, and offer much faster processing times. Fan et al.[20,21] used single laser pulses to demonstrate colours on copper, where each colour was associated with a single set of laser parameters. Plasmonic effects were mentioned as a mechanism underlying

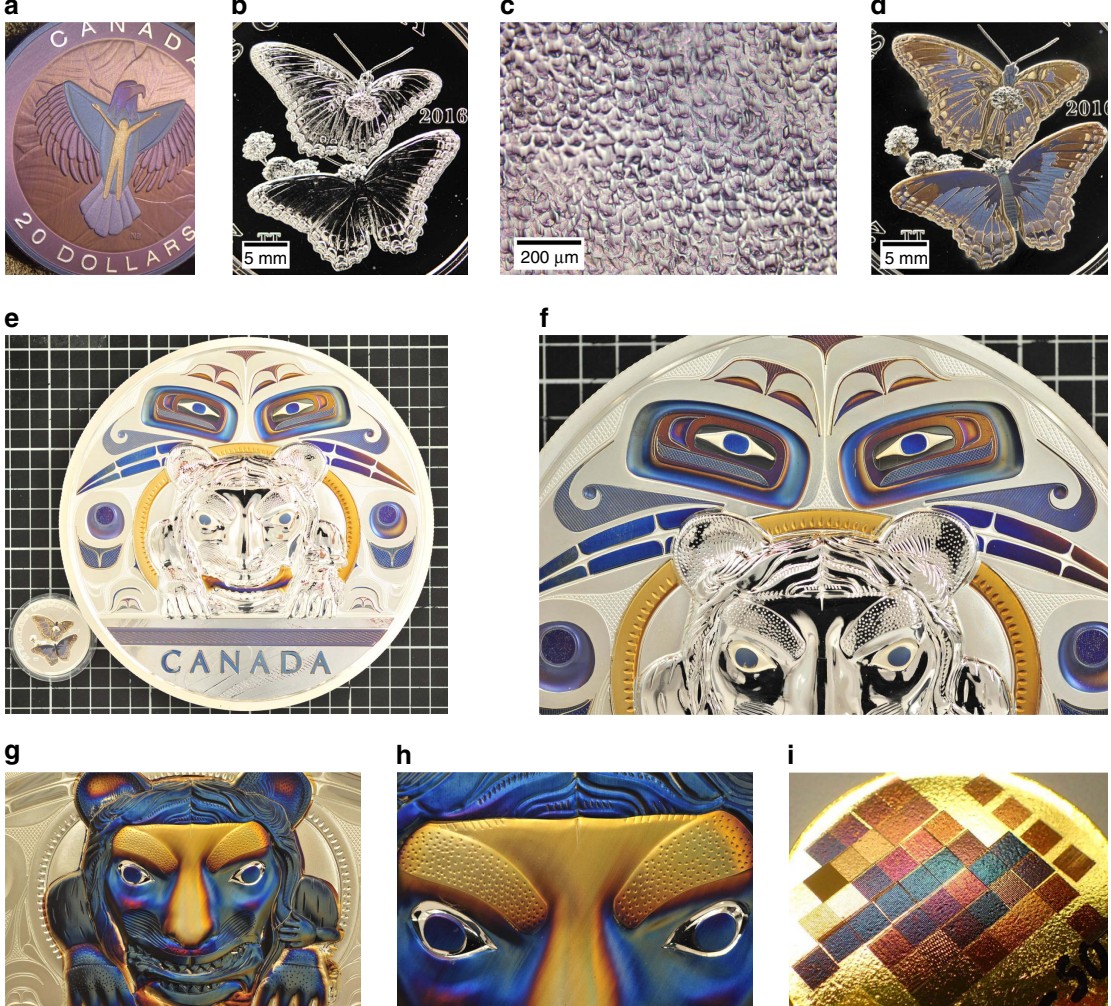

**Figure 1 | Laser colouring of large areas having significant topographic variations.** (**a**) Photograph of a coloured silver coin featuring a representation of an eagle. (**b,d**) Photographs of silver coins bearing a frosted butterfly (**b**) before and (**d**) after laser colouring. The topography on these coins has an overall height of ∼2 mm. The distinctive white areas on the butterfly are due to the frosted (that is, melted) finish; (**c**) optical microscope image of the frosted regions, having topographic variations of up to 2 mm in height and a roughness of ∼1 to 2 µm. (**e**) Photograph of a laser-coloured 5 kg silver coin of diameter 21 cm and thickness 2.5 cm (the butterfly coin of (**d**) is also shown for comparison). (**f**) Close-up of (**e**) showing crevices ∼5 mm deep of black- and white-coloured eyes. (**g**) Alternate colouring of another 5 kg coin over topographic relief ∼1 cm in height. (**h**) Close-up of (**g**) near the nose and brow, the highest points on the coin. (**i**) Angle-independent colours produced on the surface of a gold coin.

colour production, arising from NPs of medium size. Work by a different group showed that oxide production on copper could also generate colours[22].

Thermal effects for ps pulses are likely to play a role in the creation of colours, unlike for fs laser pulses that have durations that are shorter than the thermal expansion time[23]. The use of closely time-spaced laser pulses (bursts) then presents an interesting method of manipulating the laser–metal interaction. The use of laser bursts on tissue surfaces was recently shown to rapidly ablate the hot interaction region with little heat damage in a process called 'cold ablation'[24]. The use of burst for machining metals is of interest due to its increasing ablation yield and faster cutting times[25]. The advantage of burst compared with the nonburst (single pulse) regime is linked to the increase in electron–phonon coupling[26].

Here, we report a widely applicable high-throughput and deterministic process for producing a comprehensive angle-independent (that is, non-iridescent) colour palette composed of many colours via ps laser nonburst and burst colouring methods. This is achieved on unpolished metal surfaces, including silver, gold, copper and aluminium, and surfaces with frosting and centimetre-scale topographic features. We demonstrate the process by colouring silver coins produced at the Royal Canadian Mint, including large 5 kg coins. Scanning electron microscopy (SEM) images of laser-irradiated surfaces reveal the formation of random NPs having size and separation distributions that are statistically controllable through the laser parameters. In addition, we show significant increase in colour quality using the burst colouring method over the nonburst. Colour saturation (Chroma), increased by up to ∼70% in some cases, and colour lightness is observed to extend in range by almost 60%. Our experiments on nonburst demonstrate that a large set of laser parameter combinations can produce a given Hue, as long as the total accumulated fluence remains the same, making the process scalable and time efficient. To understand the colour formation, we used large-scale computational electrodynamics to simulate the scattering from periodic distributions of different sized NPs, with geometrical parameters based on statistical analyses of several SEM images of irradiated surfaces. Simulations show that plasmonic resonances in arrangements of NPs play the main role in the colour formation process. We were able to reproduce trends in experimental reflectance spectra for a range of statistical parameters, thus explaining the origin of different colours.

## Results

**Applications.** Figure 1a shows the application of our colouring process to the selective colouring of a silver coin featuring a representation of an eagle. Figure 1b,d shows images before and after colouring a silver coin displaying a butterfly having topography variations of up to 2 mm in height. The colours were selected before laser colouring using a master curve of hue versus total accumulated fluence (see discussion below) for available colour palettes. The colouring process can be applied to surfaces of varying quality, allowing, for example, the uniform colouring of rough frosted surfaces (that is, melted, Fig. 1c) or of large surfaces having significant topographic variations. For example, Fig. 1e–h shows the laser colouring of 5 kg silver coins displaying a representation of a bear, using different colour designs. These 5 kg coins are 21 cm in diameter and 2.5 cm thick, and they have topographic features as high as 1 cm above the flat regions of the surface (nose of the bear), crevices as deep as 5 mm below the flat regions of the surface (topmost eyes), and thus of overall topographic variations of ∼1.5 cm.

Precise colouring of features on coins was done using vision alignment software with pattern recognition. The vectors were obtained from an artist at the Royal Canadian Mint. The process is capable of making angle-independent colours on elevated and complex surfaces that are difficult or impossible to colour via traditional paint-based or bottom-up manufacturing processes. Moreover, colour gradients can easily be produced by focusing the laser beam slightly above the surface, outside the confocal volume, making the colouring process sensitive to topography (for example, see Fig. 1g,h). The number of colours in a colour gradient is governed by the distance between the surface and the laser focus. At small distances (while still out of the confocal region) the colour gradient has a smooth transition in colours over a large area—see stripes next to the ears of the bear in Fig. 1e,f. Flat and topographical surfaces situated within the confocal volume are, however, coloured uniformly. In the case of extreme topography, the laser energy can be increased or the focus altered to uniformly colour topographic features.

Colour palettes were also produced on gold samples, as shown in Fig. 1i, and on copper and aluminium surfaces, as shown in Supplementary Fig. 2. The creation of purple, blue and green angle-independent gold and copper is interesting, especially given the onset of vertical electronic transitions in these metals within the visible range (that is, an absorption edge), and will be the subject of further investigation.

Unprotected colours were observed to be unstable over a short timeframe. Passivation coatings formed via atomic layer deposition or evaporation on coloured silver surfaces protected the colours during aggressive humidity and tarnish tests carried out at the Royal Canadian Mint. The colours were, however, slightly red-shifted due to the passivation layer, an effect that is consistent with the shifting of plasmonic resonances on the surface. These shifts can be pre-compensated by altering the appropriate laser parameters during the initial writing.

**Colours and burst versus nonburst.** The exposure of pure silver to different laser parameters is observed to produce vivid angle-independent colours, as shown in Fig. 2. Figure 2a–c shows colour palettes obtained using our technique, specifically, with nonburst (Fig. 2a) and burst irradiation (Fig. 2b,c). Supplementary Figure 1 shows a schematic illustrating these irradiation schemes. Comprehensive colour palettes were produced through a variation of different machining techniques such as: (1) changing the line spacing, $L_s$, between each successive line (Fig. 2a); (2) changing the laser marking speed, $v$, while maintaining a fixed line spacing (Fig. 2b); (3) varying the light polarization with respect to the machining direction; and (4) passing multiple times on the same area, changing the angle between the machining direction and the light polarization, or changing the laser parameters, between each pass (for example, cross-hatching) (Fig. 2c). The same processes were used to produce colour palettes on gold, copper and aluminium.

These colours are highly reproducible and found to depend on the size and density of ablated metal NPs covering the surface (discussed further below). During laser ablation by raster scanning the sample surface, the number of metal NPs redeposited and accumulating from one line to the next is dictated by the inter-line spacing $L_s$ (or marking speed $v$); that is, the spacing between two consecutive laser lines. The change in particle density with increasing distance from the laser ablated line can be seen in Supplementary Fig. 3 (see Supplementary Note 1).

Figure 2a shows the colours produced with nonburst irradiation of silver by raster scanning the surface, where the different colours are produced by simply increasing $L_s$. Increasing $L_s$ from 1 or 2 μm in increments of 0.5 μm up to 13.5 μm, at a laser marking speed of $v = 100$ mm s$^{-1}$ results in a production rate of $\eta = 0.1$ to 1.6 mm$^2$ s$^{-1}$, where the production rate is defined as $\eta = L_s v$. Augmenting the laser marking speed in increments of

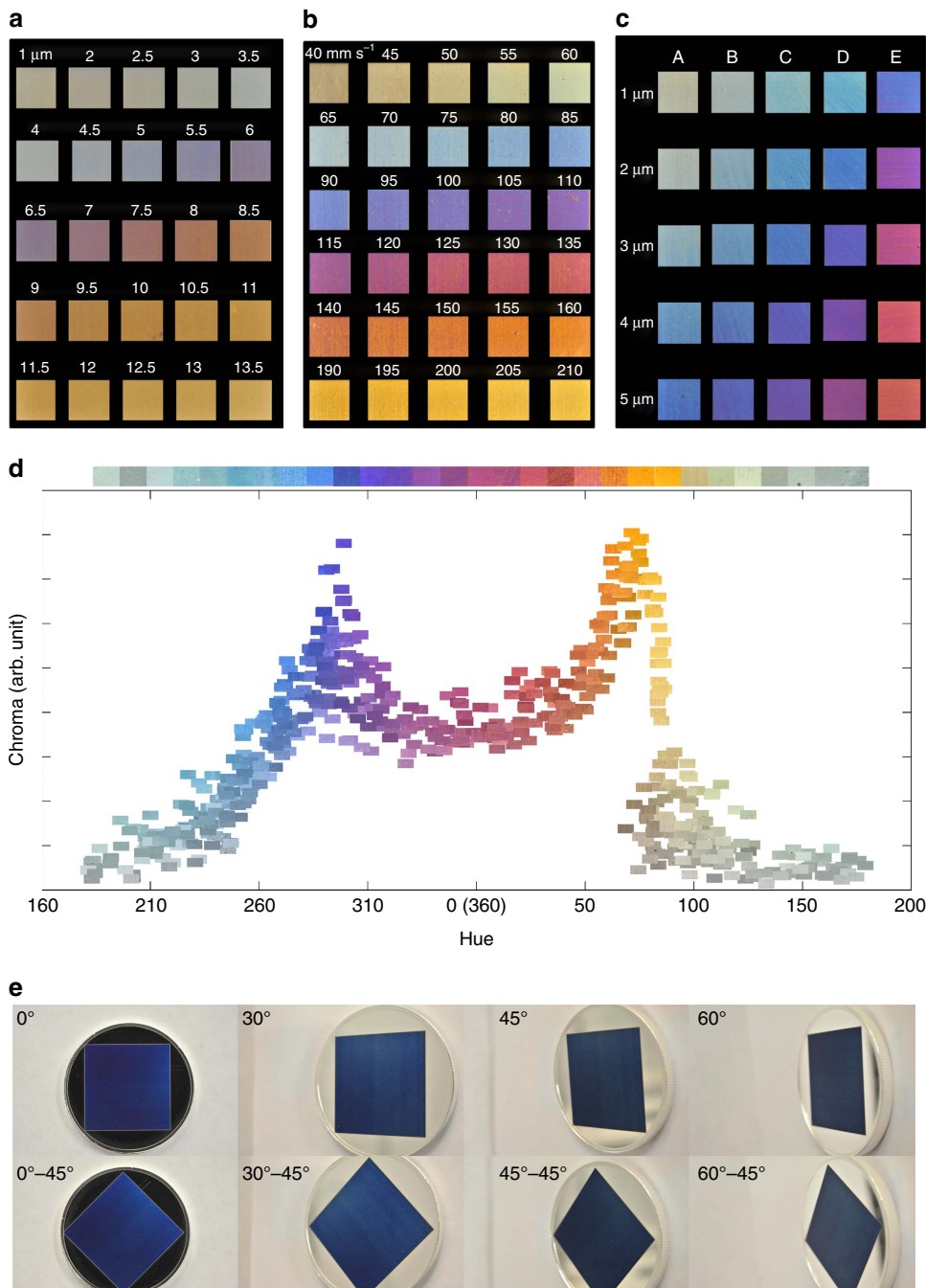

**Figure 2 | Colour palettes on Ag.** Coloured squares 25 mm$^2$ in area produced using (**a**) nonburst and (**b,c**) burst irradiation. (**a**) The different colours were obtained by changing the line spacing, $L_s$, in steps of 0.5 μm (as marked above each panel) for a laser fluence of $\varphi = 1.29$ J cm$^{-2}$ and a marking speed of $v = 100$ mm s$^{-1}$. (**b**) Colour palette obtained using bursts containing 5 laser pulses by varying the machining speed in increments of 5 mm s$^{-1}$ starting at 40 mm s$^{-1}$ (as marked above each panel) with a fixed line spacing of $L_s = 7$ μm and a laser fluence of $\varphi = 5.6$ J cm$^{-2}$. (**c**) Colour palette obtained by maintaining the line spacing fixed (row-wise, as marked) and varying the machining angle between each successive pass (that is, the angle between the light polarization and machining direction), where column A = 2 pass at 270°, column B = 1 pass at 270° + 1 pass at 234°, column C = 1 pass at 270° + 1 pass at 191°, column D = 1 pass at 270° + 1 pass at 162° and column E = 1 pass at 270° + 1 pass at 126°. The burst parameters in **c** were identical to the parameters used in **b** and the marking speed was $v = 150$ mm s$^{-1}$. (**d**) Graph of Chroma versus Hue, using as markers photographs of the colours obtained by applying the different machining processes of (**a–c**) and using the same burst parameters and fluence as in **b,c**. The horizontal colour bar was constructed from a collection of photographs of the 30 highest Chroma colours in **d**, covering the entire Hue range (0 to 360). (**e**) Photographs of a blue-coloured coin taken at four tilt angles (0º, 30º, 45º and 60º) for two coin orientations.

5 mm s$^{-1}$ from 40 to 160 mm s$^{-1}$, and from 190 to 210 mm s$^{-1}$, with $L_s = 7$ μm results in production rates of $\eta = 0.3$ to 1.5 mm$^2$ s$^{-1}$ for the colours shown in Fig. 2b. The colours shown in Fig. 2c, obtained by the cross-hatching method, maintained the line spacing fixed (as marked for each row) while varying the machining angle between each successive pass, at a laser marking speed of $v = 150$ mm s$^{-1}$ for a production rate of $\eta = 0.38$ mm$^2$ s$^{-1}$ to 0.69 mm$^2$ s$^{-1}$. The energy distribution of

each pulse within the bursts and the laser fluence were the same for both Fig. 2b,c). The energy distribution was chosen based on the quality of the colours produced. The colour palettes in Fig. 2a,c can be extended by reducing the line spacing (marking speed in Fig. 2b, or angle in Fig. 2c) between each successive line that also increases the total accumulated fluence.

Figure 2d plots Chroma versus Hue obtained from a collection of colour palettes. The colours were created using the various colouring process described above using a single fluence for the same burst energy distribution as in Fig. 2b,c). A complete Hue palette is produced, covering the entire spectral and nonspectral regions (for example, magenta). The colour bar above Fig. 2d was constructed by selecting a subset of 30 colours having the highest Chroma. Green and cyan colours have the lowest Chroma. High-quality white and black were also achieved using the appropriate laser parameters, as can be observed on the eyes in Fig. 1f.

The burst colours (Fig. 2b,c) are of higher quality, showing greater saturation than the nonburst colours (Fig. 2a). Figure 3a,b plots Chroma and Lightness versus Hue, respectively, for a large collection of colours produced using burst and nonburst irradiation using multiple laser parameters (beyond those used to produce Fig. 2d. A significant increase in Chroma values is achieved over the entire Hue range using burst irradiation compared with nonburst; for example, the Chroma of Hue values near 285 is increased by ∼70% (Fig. 3a). Similarly, the range of Lightness values is also increased, for example, by ∼60% for Hue values near 100 (Fig. 3b). Fig. 3c shows a CIE *xy* chromaticity diagram of the colours plotted in Fig. 3a,b) produced using burst and nonburst irradiation. The range of colours achievable is significantly larger using burst irradiation, and the range is expected to increase further as more work using this approach is pursued. The enhancement of colours using burst was also observed in the creation of colour palettes on gold, copper and aluminium.

Our chromaticity diagram on silver (Fig. 3c) is comparable to the chromaticity diagram obtained by bottom-up lithographic techniques using aluminium[27–29] with the exception of the green colours. The creation of extended colour palettes on aluminium was achieved by tuning the interspacing of fabricated bimodal nanostructures[11]—interestingly, we find that laser-induced bimodal distributions of NPs are essential to the creation of a broad colour palette, as discussed in the forthcoming sections. An advantage of bottom-up techniques to render plasmonic colours is their ability to achieve high resolution by defining nanoscale features[8,10]. Our top-down technique holds advantages in process simplicity (single-step direct laser write), and in that no cleanroom infrastructure, moulds, stamping die or chemical processing are required. The existence of a master curve allows colouring processes that tradeoff laser fluence against writing speed. Our technique is capable of colouring large areas bearing high topographic relief (Fig. 1) that may be challenging to work with using bottom-up techniques or paint application methods (for example, ink-jet spray over large areas when encountering sharp relief).

**Nonburst colouring of silver and total accumulated fluence.** We focus now on the nonburst colouring process applied to silver to build an understanding of the nanostructures induced on irradiated surfaces and of their ability to render colour. For this purpose, colours were produced using nonburst irradiation at a fixed repetition rate of 50 kHz, keeping the light polarization parallel to the laser machining direction. Colour palettes were produced following the approach of Fig. 2a where $L_s$ was changed in fixed intervals. During production of the colour palettes, it was

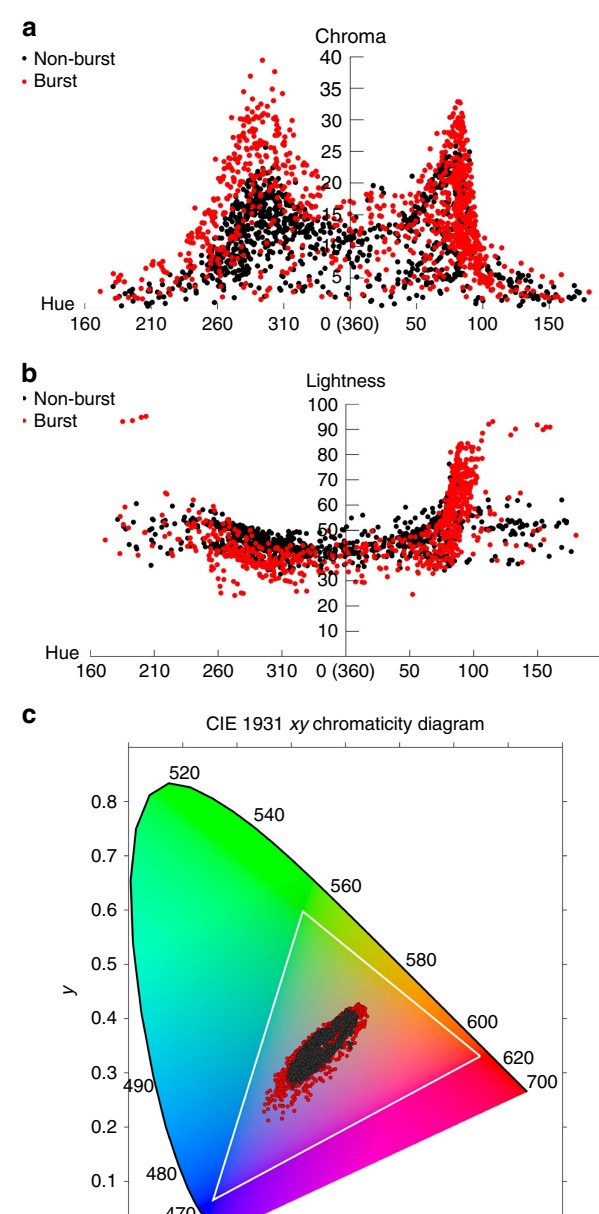

**Figure 3 | Nonburst versus burst colours.** Graphs comparing (**a**) Chroma and (**b**) Lightness values versus Hue for nonburst and burst produced colours. The burst colouring method is observed to significantly increase the Chroma of each Hue value by up to ∼70% compared with the nonburst colouring method. (**c**) CIE *xy* Chromaticity diagram comparing the burst (red dots) and nonburst (black dots) colours. The white triangle shows the gamut of the sRGB colour space, representing colours achievable by common computer monitors.

noticed that similar Hue values were often obtained using different laser parameters. Each Hue value was subsequently found to be linked to a unique total accumulated fluence value, as plotted in Fig. 4a.

The total accumulated fluence Φ is defined as:

$$\Phi = \varphi N_{eff} = \frac{a^2 E f}{v L_s} \qquad (1)$$

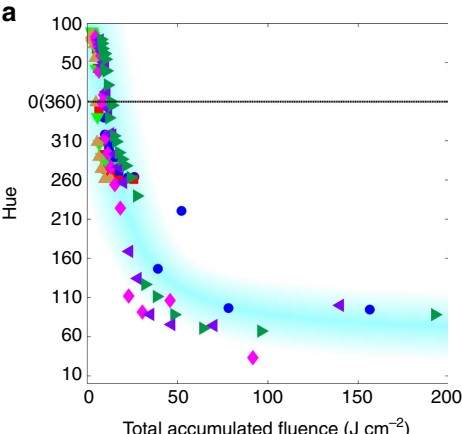

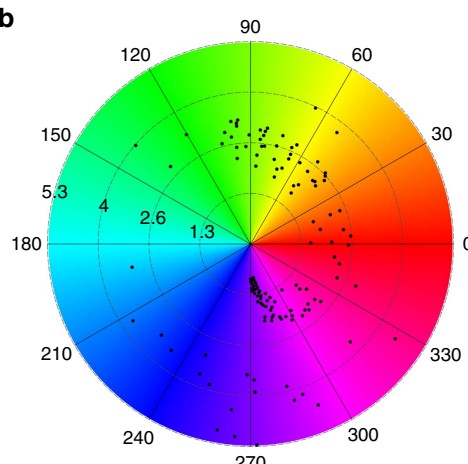

**Figure 4 | Hue versus total accumulated fluence.** (**a**) Hue versus total accumulated fluence for (blue circle) $\varphi = 1.12\,\mathrm{J\,cm^{-2}}$ at $v = 11\,\mathrm{mm\,s^{-1}}$, (red star) $\varphi = 1.67\,\mathrm{J\,cm^{-2}}$ at $v = 50\,\mathrm{mm\,s^{-1}}$, (green downward pointing triangle) $\varphi = 2.59\,\mathrm{J\,cm^{-2}}$ at $v = 100\,\mathrm{mm\,s^{-1}}$, (brown left pointing triangle) $\varphi = 6.26\,\mathrm{J\,cm^{-2}}$ at $v = 250\,\mathrm{mm\,s^{-1}}$, (pink asterisk) $\varphi = 13.23\,\mathrm{J\,cm^{-2}}$ at $v = 500\,\mathrm{mm\,s^{-1}}$, (purple left pointing triangle) $\varphi = 20.58\,\mathrm{J\,cm^{-2}}$ at $v = 750\,\mathrm{mm\,s^{-1}}$ and (green right pointing triangle) $\varphi = 29.75\,\mathrm{J\,cm^{-2}}$ at $v = 1{,}000\,\mathrm{mm\,s^{-1}}$. The Hue value completes a full 360° rotation. (**b**) Polar plot representation of (**a**) with Hue plotted azimuthally and the logarithm of the total accumulated fluence plotted radially. A full rotation in Hue is observed with increasing total accumulated fluence.

where the laser fluence $\varphi$ is given by:

$$\varphi = \frac{2E}{\pi\omega_o^2} \qquad (2)$$

and the effective number of laser shots $N_{\mathrm{eff}}$ is

$$N_{\mathrm{eff}} = \sqrt{\frac{\pi}{2}\frac{a\omega_o f}{v}}\sqrt{\frac{\pi}{2}\frac{a\omega_o}{L_s}}. \qquad (3)$$

where $E$ is the laser pulse energy, $\omega_o$ the beam waist radius, $f$ the laser repetition rate and $a$ a correction factor due to the modification of a larger area when using a higher pulse energy. This correction factor was obtained from semi-logarithmic plots used to determine the laser spot size, following the procedure detailed in ref. 30. The intraline component of $N_{\mathrm{eff}}$ encompasses the shot overlap within a local region, where $f/v$ is the distance travelled between successive laser pulses in a single laser line. The interline component, in comparison, considers the geometrical overlap, $a\omega_o/L_s$, between successive laser lines. At large spacings,

that is, for a lower total accumulated fluence and as $a\omega_o/L_s \to 1$, the colours converge to yellow due to the absence of overlap between consecutive laser lines. Further evidence of this accumulation process can be observed from the absence of colours, other than yellow, in the last line of each of the coloured squares (not shown). The last line does not undergo the process of particle accumulation as there are no subsequent lasered lines.

We find that a very large number of laser parameter combinations can be used to obtain a given Hue, as long as the total accumulated fluence remains the same, under the condition that the repetition rate remains fixed—this is highlighted by the light blue trend band (master curve) on Fig. 4a. Figure 4b shows a polar plot of Fig. 4a with Hue plotted azimuthally and the logarithm of the total accumulated fluence plotted radially; a full 360° rotation in Hue is observed.

The master curve is useful, for example, to tradeoff writing time versus laser write parameters; that is, from equation (1), $v$, $L_s$ and $E$ can be changed independently to obtain a specific total accumulated fluence and the master curve used to relate the fluence to the Hue. The largest production rates attained in producing a comprehensive colour palette from the master curve were in the range from $\eta = 3\,\mathrm{mm^2\,s^{-1}}$ to $36\,\mathrm{mm^2\,s^{-1}}$, achieved by setting $v = 3{,}000\,\mathrm{mm\,s^{-1}}$ (the highest speed of our galvometric mirrors). The master curve can also assist with the aesthetic control of a single colour, as the Lightness increases with laser fluence, $\varphi$, for a constant Hue; for example, various 'types' of blue (navy blue, sky blue) can be realized in this manner. Chroma, however, was observed to remain unaffected for each Hue when using different laser fluences, $\varphi$. The rotation and overlapping of the data points on Fig. 4b show explicitly that the $xy$ values of CIE XYZ colour space (Fig. 3c) can be recovered for a fixed laser fluence, $\varphi$, by simply adjusting the marking speed or line spacing.

Deviations from the master curve in Fig. 4a are attributed to the increasingly chaotic nature of the surface with increasing total accumulated fluence. Also, while we are able to render reds reproducibly, there is less variety in the reds than for other colours due to the high slope in the red region on the master curve (Hue values from 345 to 15—modulo 360°). In fact, the rate of change of Hue per μm (or mm s$^{-1}$), in the red region, is twice as high as the rate of change in all other Hue regions, explaining the absence of red colours in Fig. 2a.

Changing the repetition rate, $f$, along with $v$, $L_s$ and $E$ to obtain a desired total accumulated fluence resulted in colours other than what was expected based on the master curve of Fig. 4a. To study the effect of the laser repetition rate on the colours, $f$ was changed proportionally with speed, $v$, to maintain a fixed total accumulated fluence, while $E$ and $L_s$ were kept constant. The colour palette obtained was similar to that shown in Fig. 2a but with fewer colours. Increasing $f$ and keeping it fixed while changing $v$, $L_s$ and $E$ would only create reduced colour palettes. The baseline in Fig. 4a increased with increasing repetition rate gradually cutting off the lower colours (that is, blues, purples and reds). No colours other than yellow were observed at repetition rates above 400 kHz, suggesting a local thermal accumulation effect.

**Surface analysis.** Extensive SEM analyses of regions exhibiting different colours on silver reveal 3 distinct classes of particles differentiated by size: large ($R \geq 75\,\mathrm{nm}$), medium ($10.7 \leq R < 75\,\mathrm{nm}$) and small ($R < 10.7\,\mathrm{nm}$), where $R$ is the radius of a particle. These particle classes were obtained from statistical analyses of SEM images, producing histograms with well-defined ranges of particle sizes (see Supplementary Figs 4 and 5).

Figure 5a–c shows three coloured surfaces with corresponding low- and high-magnification SEM images. The particle density is

found to change significantly with line spacing, as shown in Fig. 5. In Fig. 5d–i, the small and medium particles are seen to form random networks with the medium-sized particles sparsely covering the surface and the small ones more densely distributed across the irradiated region (see Supplementary Note 2). The formation of NPs is believed to come from the combination of thermal effects[31–34] and the redeposition of particles following laser ablation[17,15]. Upon close examination of the SEM images, it appears that the small particles are in reality approximately hemispherical, an observation supported by a cross-section of a coloured silver sample obtained through focused ion beam milling and imaging. Thus, they can be modelled as spheres partially embedded into the silver surface.

SEM analyses of coloured regions reveal that the number density of the small particles, produced using different laser parameters, follows its own distinctive trend with total accumulated fluence (see Supplementary Fig. 4a), similar to that of Fig. 4a, whereas the number density of the medium particles does not (see Supplementary Fig. 4b). This observation suggests that the small particles play a role in the colours perceived. The mean radius of small and medium particles was found to remain approximately constant as a function of line spacing (Supplementary Fig. 4c,d, determined from the analysis of three SEM images per line spacing). However, the mean interparticle distance (wall-to-wall) changes with line spacing (Supplementary Fig. 5e,f)), that is, it decreases for small particles and increases for medium particles, suggesting that the colours are affected by near-field interactions between nanoparticles in close proximity[6,35–37], particularly the associated surface plasmon resonance[6].

Wavelength-dispersive spectroscopy analysis of the different colours in Figs 2a and 4a showed no difference in the amount of oxidation measuring $2.8 \pm 0.4\%$ oxygen content for all Hue values tested. Monte Carlo simulations (WinXray) of silver oxide layers, under the same wavelength-dispersive spectroscopy operating conditions, gave an equivalent oxide thickness of $\sim 2$ nm. The same conclusion was gathered from the invariability of the oxygen content in separate energy-dispersive spectroscopy and X-ray photoelectron spectroscopy measurements of the different coloured surfaces. The X-ray photoelectron spectroscopy analyses of the coloured surfaces also showed no sulfur content.

**FDTD simulations**. Our finite-difference time-domain (FDTD) simulations show that colour formation can be viewed as spectrally selective absorption processes involving plasmonic resonances. White light incident on a nanostructured surface is not fully reflected, as it would be for a smooth silver surface. Rather, narrow-band spectral components (colours) are subtracted due to plasmonic resonances on the surface, and absorption processes in the medium, as also reported in another study[38]. What survives these absorptive processes is reflected (there is no transmittance).

In Fig. 5j we show a high-magnification SEM image for the case of $\varphi = 1.67$ J cm$^{-2}$ at $v = 50$ mm s$^{-1}$ with $L_s = 8$ µm, from which we observe the presence of NPs having a random distribution. The analysis of SEM images produced statistics of the NP distributions, that is, number of particles per unit area, particle dimensions and interparticle distances, from which the average radii and average interparticle distances were obtained. Three sets of NPs were identified based on their dimensions, that is, small, medium and large NPs. The large particles are neglected in our analysis due to their low number on the surface, as suggested by the bimodal distribution for small and medium NPs in the histograms (Supplementary Fig. 5a–c). We carried out simulations considering only small and medium Ag NPs uniformly distributed on a perfectly flat Ag substrate following the statistics retrieved from the SEM images. The average radii of small and medium NPs used in simulations are denoted $R_s$ and $R_m$, and the average interparticle distances (centre-to-centre) are denoted $D_s$ and $D_m$, respectively.

Images such as Fig. 5j inspired us to construct a model of surfaces where NPs were periodically arranged in a hexagonal configuration that produces a hexamer unit cell. The hexamer has $D_{6h}$ symmetry, and based on group theory, exhibits polarization independence (isotropy), as verified by simulations[39]. We chose $D_m$ as an integer multiple of $D_s$ to allow the application of periodic boundary conditions to the unit cell defined by the medium NPs. The periodicity of the unit cell ($\sim D_m$) is small enough to preclude coupling by diffraction into propagating surface plasmon waves on the Ag surface. Perfect periodicity of the NPs gives isotropy with respect to the incident polarization, narrowband resonances and ultimately colour selectivity. This can be considered a good qualitative approximation to actual surfaces.

By changing the surface density and/or the radius of small and/or medium NPs, several types of arrangements can be formed. An arrangement is homogeneous when it is formed by an aggregate of small NPs only or medium NPs only, and heterogeneous when it is formed by an aggregate of medium and small NPs. A heterogeneous arrangement, for example, is formed by a central medium NP with nearby small NPs arranged in a ring. Furthermore, by modifying the level of embedding of the NPs into the substrate, the geometry changes because the NPs are gradually transformed from spheres to hemispheres. Plasmonic arrangements embedded in a homogeneous medium or on a dielectric have been extensively studied (dimer, trimer, quadrumer, tetramer, hexamer, heptamer)[39–41], but not on a metal substrate and not arranged and embedded as inspired by SEM images of laser-processed metal surfaces. We have observed different types of resonant modes in our simulation of arrangements; we will refer to them as collective resonances.

In Fig. 6a,b we demonstrate the interplay between medium and small NPs on colour formation, and the effect of embedding the small NPs on the collective resonance, respectively, by showing simulation results for the statistical average parameters associated with the line spacing $L_s = 5$ µm, marked at $v = 50$ mm s$^{-1}$ with a laser fluence $\varphi = 1.67$ J cm$^{-2}$: $R_s = 4$ nm, $R_m = 34.3$ nm, $D_s = 13.5$ nm and $D_m = 108$ nm.

In Fig. 6a we show the computed reflectance spectrum considering small NPs only, medium NPs only and the combination of both NP sizes; the small and medium NPs are embedded by half their radius. We observe the resonance of the medium-sized NPs at $\lambda = 390$ nm (black curve), shifted slightly from the prediction of Mie theory for the same sphere in air, due to the presence of the substrate. The interaction between small and medium NPs drastically alters the computed reflection spectrum, as noted by the appearance of a deep dip at $\lambda = 650$ nm in Fig. 6a (red curve) that is due to a new collective resonance. In Fig. 6b we show how the reflectance response evolves as the small NPs are embedded in steps of 0.5 nm. A very high sensitivity of collective resonances on embedding is observed as the large shift in the associated dip. By embedding the small NPs, the size of the nanogaps between the central medium NP and the surrounding small NPs increases. This changes dramatically the geometry of the arrangement and, consequently, their resonances. A blue shift in the collective resonance is observed with increasing embedding (a blue shift due to cluster expansion was reported in previous work[39]). In the laser-generated surfaces, we would expect the location and embedding of the small NPs to vary across the surface, and thus the overall effect of the sharp absorption dip to be averaged out; we take this into account below. In Fig. 6b we also note other smaller features in the reflectance curves. These are due to resonances arising from homogeneous arrangements of small NPs only.

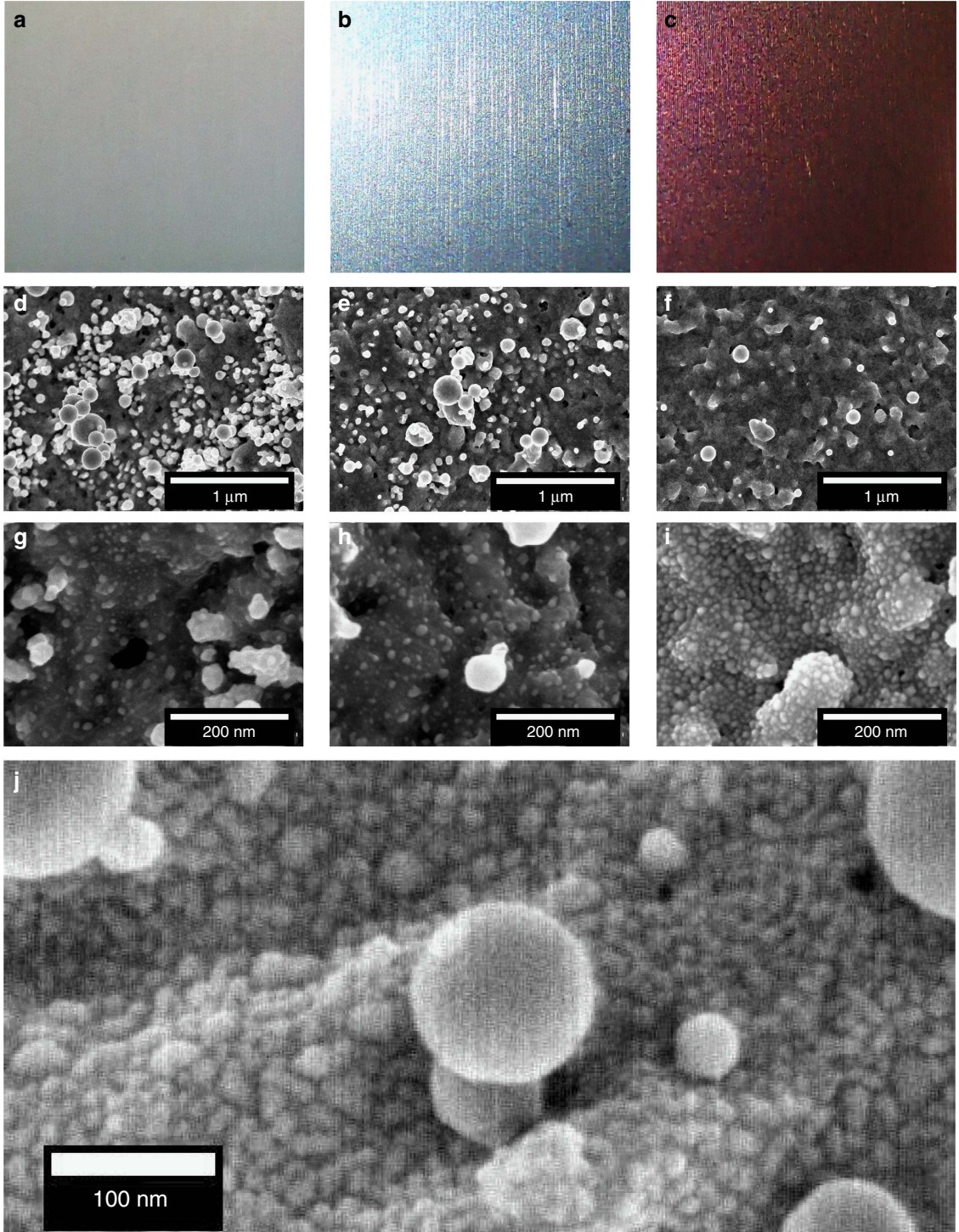

**Figure 5 | Nanoparticle networks on coloured silver surfaces.** (**a–c**) Low-magnification optical microscope images of coloured surfaces; (**d–f**) low-magnification and (**g–i**) high-magnification SEM images of corresponding surfaces. Surfaces processed using $\varphi = 1.12 \, \text{J cm}^{-2}$ at $v = 11 \, \text{mm s}^{-1}$. (**a,d,g**) $L_s = 5 \, \mu\text{m}$ (Hue = 216.5, Cyan); (**b,e,h**) $L_s = 10 \, \mu\text{m}$ (Hue = 269.3, Blue); and (**c,f,i**) $L_s = 30 \, \mu\text{m}$ (Hue = 17.2, Red). The number of medium NPs is observed to decrease with increasing $L_s$ (**d,e,f**, scale bar, 1 μm), and the number of small NPs to increase with $L_s$ (**g,h,i**, scale bar, 200 nm). (**j**) High-magnification SEM image for $\varphi = 1.67 \, \text{J cm}^{-2}$ at $v = 50 \, \text{mm s}^{-1}$ with $L_s = 8 \, \mu\text{m}$.

In Fig. 7a–d we show the electric field distribution at the free-space optical wavelengths of $\lambda = 390 \, \text{nm}$ (Fig. 7a,c) and $\lambda = 650 \, \text{nm}$ (Fig. 7b,d), where we observe absorption dips in Fig. 6a. We show $xz$ planes cut 2 nm above the silver surface for medium NPs only (Fig. 7a,b), and for medium and small NPs (Fig. 7c,d). In Fig. 7a we show the response of the surface to illumination with $\lambda = 390 \, \text{nm}$, the resonance of the medium-sized NPs only.

Figure 7b shows that medium NPs only do not produce any absorption when illuminated with $\lambda = 650 \, \text{nm}$, and where the reflectance in Fig. 6a is shown to be very high. In Fig. 7c we observe that the near-field intensity of the medium NPs at $\lambda = 390 \, \text{nm}$ is reduced by the presence of the small NPs that act as plasmonic waveguides[42], coupling the medium-size NPs. When small and medium NPs are illuminated at $\lambda = 650 \, \text{nm}$ (Fig. 7d), a

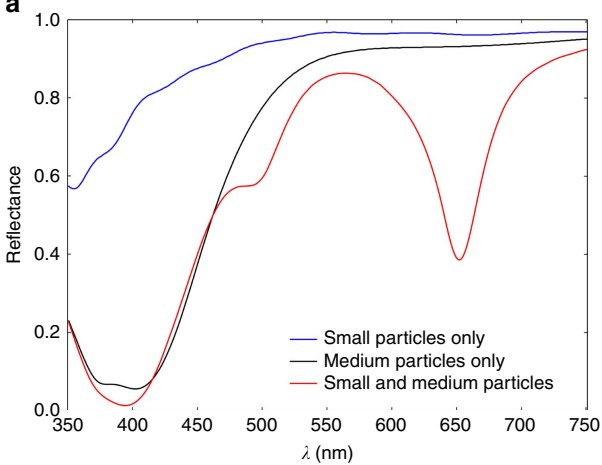

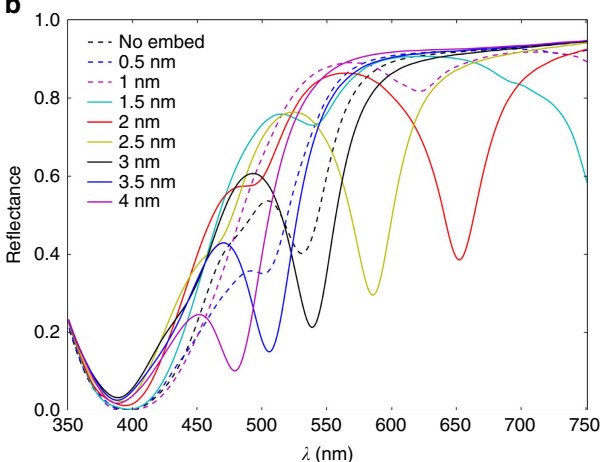

**Figure 6 | Computed reflectance responses.** (**a**) Computed reflectance spectra considering only small NPs embedded by $R_s/2$; only medium NPs embedded by $R_m/2$; and medium and small NPs embedded by $R_m/2$ and $R_s/2$, respectively. (**b**) Reflectance spectra computed by varying the embedding of the small NPs in steps of 0.5 nm (the medium NPs are embedded by $R_m/2$).

collective resonance is excited, producing field enhancement in the nanogaps, and ultimately the smaller absorption dip observed in Fig. 6a. The interaction of a $z$-polarized and $y$-propagating plane wave pulse with the nanoparticles on the surface (medium and small NPs are embedded into the surface by $R_m/2$ and $R_s/2$, respectively) is shown in Supplementary Movies 1 and 2, for an $xz$ plane 2 nm above the silver substrate, and a $yz$ plane through the middle of the medium particles, respectively. In Fig. 7e we show a snapshot extracted from Supplementary Movie 2; the plane wave excitation pulse has just hit the nanostructures, inducing localized surface plasmons in all NPs that are clearly visible.

In Fig. 8a we show measured reflectance spectra of 6 coloured squares produced with line spacings of 4, 5, 6, 8, 10 and 14 μm. These reflectance spectra show a transition from blue to yellow with increasing line spacing. The same transition has already been shown in Fig. 4a as a function of the total accumulated fluence. Magenta and yellow are the colours that can be produced with the highest Chroma (note the bimodal distribution of Fig. 3a).

In Fig. 8b we reproduce the qualitative features of Fig. 8a by modelling the response of two-particle periodic surfaces according to the statistical trends observed from the corresponding SEM

images. Exact quantitative comparisons are not possible, as the experiments are on rough surfaces (for example, burnished then laser-coloured surfaces), whereas we consider a flat surface with NPs in our simulations. This also explains why the overall reflectance levels in the experiments are lower. To account for the statistical nature of embedding, we repeated the simulations for small NP embedding values ranging between 0.5 and 3.5 nm in 0.5 nm increments; the medium NPs were all embedded by 30% of their radius. We averaged over the resulting reflectance spectra to produce each curve in Fig. 8b. Averaging over nm scale embedding washes out the effect of the collective medium/small NP resonances, resulting in a broadened response, consistent with what was observed in random clusters[43]. The Hues in Fig. 8b of 276, 328, 14, 55, 65, 81 and 87 were obtained for $D_m = 81$, 86, 90, 96, 100, 113 and 139 nm, and $D_m/D_s$ ratios of 5, 5, 6, 7, 8, 10 and 13, respectively, with $R_m = 36$ nm and $R_s = 4$ nm. As in the experimental reflection spectra, a transition from blue to yellow occurs with increasing $D_m$ and decreasing $D_s$, supported by the statistical analysis of the coloured surfaces (see Supplementary Fig. 4).

Simulations show that blue originates when medium NPs are very close to each other (edge-to-edge distance $\sim 10$ nm), and yellow originates when medium NPs are farther apart ($D_m > \sim 110$ nm). This dependence on interparticle distance is in agreement with experimental trends. However, we use somewhat smaller values of $D_m$ compared with those reported in Supplementary Fig. 4. Since the statistical SEM analysis averages $D_m$ over a large surface, the interparticle distances information for very close medium NPs (which are visible, for example, in Fig. 5) is lost. This suggests that arrangements of very closely spaced medium NPs play the main role in the blue/violet colour formation, acting as colour hot spots on the surface. The colours in between blue and yellow are created with small changes in $D_m$, and thus harder to obtain experimentally with a random distribution of nanoparticles. Thus, we would expect that it would be more difficult to get a red with high Chroma, and this is precisely what is noted in experiments (Figs 2d and 3a).

In conclusion, we described a broadly applicable and deterministic process for the non-iridescent colouring of metal surfaces and demonstrated the process by colouring silver coins produced at the Royal Canadian Mint, as well as gold, copper and aluminium surfaces. In the case of nonburst irradiation, each individual colour can be linked to a total accumulated fluence that can then be exploited to decrease the colouring time of large metal surfaces or to fine-tune the colour perceived for a specific Hue. The colours originate from random distributions of small and medium nanoparticles embedded into the surface, induced and controlled by laser exposure. The randomness of the nanoparticle networks is modelled effectively by assuming a periodic structure defined by statistical averages of nanoparticle size and separation. We have demonstrated that plasmonic effects arising in nanoparticle arrangements explain the palette of experimental colours. The medium nanoparticles in particular play a fundamental role in the colour formation, as their interparticle distance modifies plasmonic resonance conditions and then the perceived colour. Irradiation using short time-spaced laser bursts was shown to increase significantly the Chroma for all Hue values, in some cases by up to $\sim 70\%$. Furthermore, burst irradiation allowed for the creation of colour palettes on surfaces of gold, copper and aluminium. This laser colouring process was demonstrated on several silver coins, including a large 5 kg coin bearing significant topographic features ($\sim 1.5$ cm). The proposed method opens the door to large-scale industrial applications of laser colouring for anti-counterfeiting, biosensing, biocompatibility and the decoration of consumer products such as jewels, art, architectural elements and fashion items.

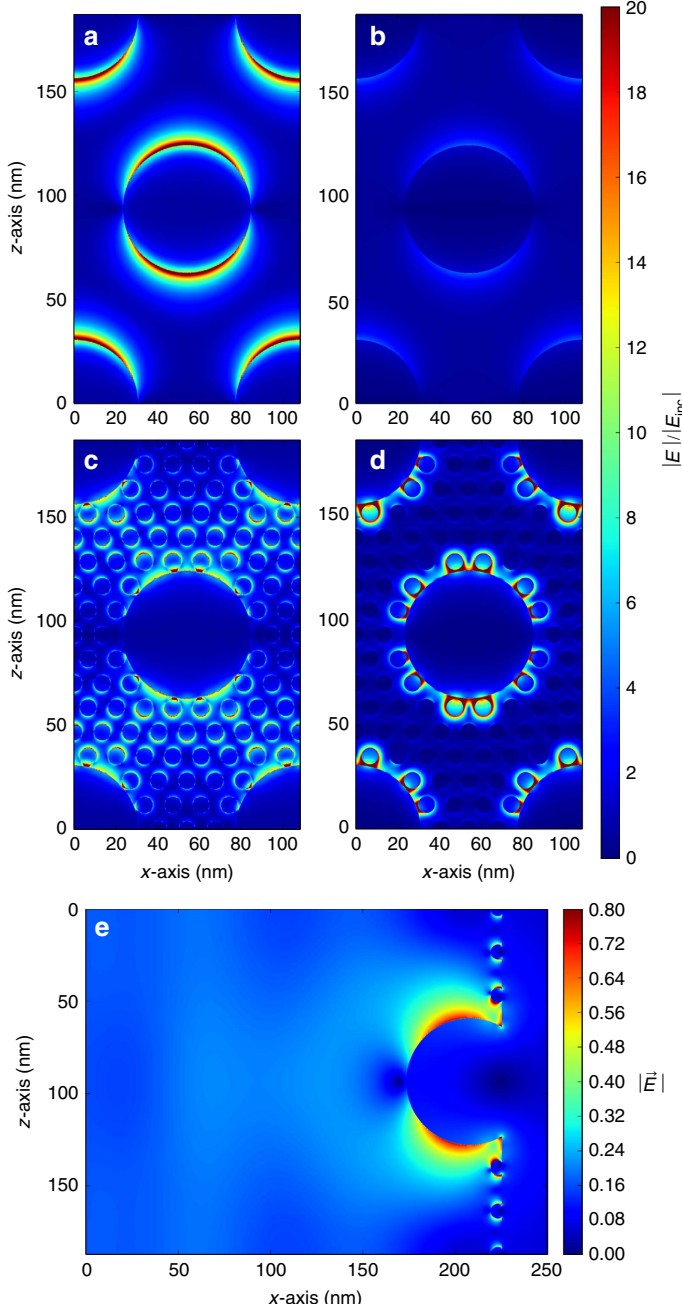

**Figure 7 | Computed near-field distributions for arrays of nanoparticles.** FDTD simulations showing the electric field distribution for only medium NPs at (**a**) $\lambda = 390$ nm and (**b**) $\lambda = 650$ nm, and for medium and small NPs at (**c**) $\lambda = 390$ nm and (**d**) $\lambda = 650$ nm. (**e**) Snapshot from the time-domain simulation for medium and small NPs. All NPs are embedded into the surface by half of their radius.

## Methods

**Laser specifications.** In our experiments, we used 1,064 nm light from a 15 W Duetto (Nd:YVO$_4$, Time-Bandwidth Product) mode-locked MOPA laser, operating at a fixed burst repetition rate of $f_b = 1/T_b = 50$ kHz ($T_b$ is the burst period) of 10 ps pulses. The time separation between each pulse within a burst is $t_{ib} = 12.8$ ns (see also Supplementary Fig. 1). A selection of 1 to 8 pulses within each burst is available. The setting of 1 pulse per burst corresponds to the normal mode of operation of the laser and will be defined as nonburst (that is, a burst of 1 pulse). The energy distribution of each pulse within a burst was controlled using the software provided by the manufacturer (FlexBurst). A spot size of 14 μm for the 163 mm lens (28 μm for the 254 mm lens) was obtained from semilogarithmic plots of the square diameter of the modified region, measured with SEM, as a function of energy, following the procedure described in ref. 30.

**Laser colouring process.** The light was focused on the metal surface using an F-theta lens ($f = 163$ and 254 mm, Rodenstock). The pulse energy for the results

presented ranged between 3.4 and 91.4 μJ. The laser was fully electronically integrated and enclosed by a third party for industrial applications (GPC-PSL, FOBA). For accurate focusing, the surface of a sample was located using a touch probe system. The silver and gold samples were of 99.99% and 99.999% purity, respectively, and not polished before machining to meet requirements of reproducibility in industrial applications. The aluminium samples were from the 7000 alloy series. For machining, a sample was placed on a 3-axis translation stage with a resolution of 1 μm in both the lateral and axial directions. Samples were raster scanned using galvanometric XY mirrors (Turboscan 10, Raylase) displacing the beam in a top to bottom fashion with a mechanical shutter blocking the beam between successive lines. The laser power was computer controlled via a laser interface and calibrated using a power meter (3A-P-QUAD, OPHIR).

**Surface imaging and statistics.** High-resolution SEM (JSM-7500F FESEM, JEOL) images were obtained using secondary electron imaging mode. For analysis of the SEM images, a Matlab program was written to locate the position of each

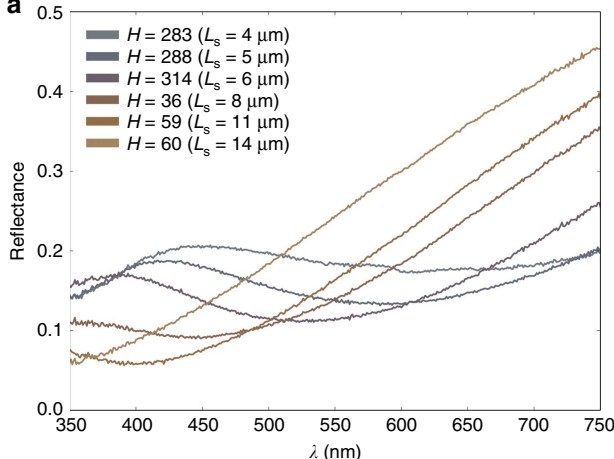

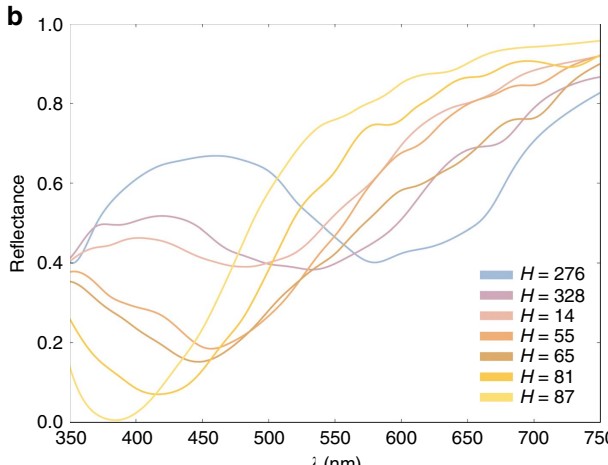

**Figure 8 | Experimental and theoretical reflectance spectra.**
(**a**) Measured reflectance spectra of colours produced with the line spacings of 4 (Hue = 283), 5 (Hue = 288), 6 (Hue = 314), 8 (Hue = 36), 10 (Hue = 59) and 14 μm (Hue = 60). (**b**) Computed reflectance spectra, each curve averaged over a few values of small NP embedding for increasing $D_m$ (Hue). The colour scheme of the curves follows the measured/calculated Hues.

particle and record its diameter and the average wall-to-wall inter-distance spacing to its nearest neighbours.

**Photometric characterization.** Colours were quantified using a Chroma meter (CR-241, Konica Minolta) in the CIELCH colour space, 2 observer and illuminant C (North sky daylight); where L is colour Lightness, C is Chroma (colour saturation) and H is Hue (colour value associated with a 360° polar scale).

**Radiometric characterization.** The reflectance measurements were performed using a CARY 7000 UV-Vis-NIR spectrophotometer (Agilent Technologies) equipped with an integrating sphere detector (Labsphere) for collecting simultaneously the specular and diffuse reflectance signals from the samples. An aperture was used in front of the samples to ensure that only the treated surface area was probed with the incident light beam. The reflectance data were calculated taking into account this aperture, and corrected against reference samples of silicon and silver.

**Numerical simulations.** Three-dimensional FDTD simulations[44,45] were performed to determine the origin of the colour rendering process. We used in-house 3D-FDTD parallel code[46,47] on an IBM BlueGene/Q supercomputer (64k cores) part of the Southern Ontario Smart Computing Innovation Platform (SOSCIP). In the model, metal NPs were arranged in the xz plane, as inspired by SEM images of irradiated surfaces and the corresponding statistics of the metal NP distributions, and the system was excited by a z-polarized plane wave. A broadband electromagnetic pulse propagating along the y-direction from air and impinging on the nanostructured surface was used. The analysis was performed over the

wavelength range 350-750 nm in a single run of the code using in-line discrete Fourier transform. Space-steps in the range of 0.125–0.5 nm were used for the simulations. The dispersion of silver was introduced by the Drude + 2CP model[48]. This model was implemented in FDTD by the auxiliary differential equation technique[49]. The simulation domain in the direction of propagation of the plane wave was truncated by convolutional perfectly matched layer absorbing boundary conditions[50]. The simulations required up to 16k cores. The theoretical reflectance spectrum was calculated by integration of the Poynting vector in the backward far-field region. The theoretical reflectance spectra were reconstructed as colours using in-house Matlab code, weighting each frequency composing the spectra to the spectral sensitivity of the human eye.

**Data availability.** The data that support the findings presented in this paper are available. Restrictions apply to data availability due to the commercial nature of the technology and are not publically available. Data are however available from the authors on reasonable request and with permission from the Royal Canadian Mint.

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

## Acknowledgements

We acknowledge the Royal Canadian Mint, the Natural Sciences and Engineering Council of Canada, the Canada Research Chairs program, the Southern Ontario Smart Computing Innovation Platform (SOSCIP) and SciNet. We acknowledge Alessandro Vaccari at Fondazione Bruno Kessler (Italy), and Graham Killaire, Josh Baxter and Meagan Ginn, COOP students at the University of Ottawa.

## Author contributions

J.-M.G. conceived the laser colouring process. J.-M.G. and G.C. developed the laser colouring technology. A.W. proposed and directed the project, and supervised the experiments. L.R. proposed the computational study. L.R. and P.B. co-directed the study and supervised the simulations. J.-M.G., G.C. and M.C. carried out the statistical analysis of the surfaces. G.C. wrote the Matlab codes. D.P. obtained reflectance measurements on coloured surfaces. A.C.L. set up the FDTD code, conducted the simulations and generated the movies. A.C.L. and J.-M.G. analysed the results. A.C.L., L.R. and P.B. formulated the theoretical conclusions. J.-M.G., A.C.L., L.R., P.B. and A.W. co-wrote the paper and J.-M.G., A.C.L. and G.C. prepared the figures. All authors discussed the experimental and theoretical results.

## Additional information

**Competing interests:** The authors declare no competing financial interests.

