## [Peer Review File · Nature Communications]

Reviewers' comments:

Reviewer #1 (Remarks to the Author):

The authors have used ps laser to make nanoparticles on silver surface so that different visible colors can be realized due to the light absorption of nanoparticles with different sizes on the surface. First, as the authors agreed the original idea of making such color pattern on metal surface is not new, as the references cited in the manuscript, Refs. 18-20. Instead, the authors just try to tune the set of laser parameters of the laser machining to get more colors. The whole manuscript is written in a way to be too technical but not scientific enough. It discusses a lot of technical part, such as what is the ps laser parameters and lots of experimental characterization details. Also although the authors claimed that their method is "producing a complete angle-independent colour palette composed of thousands of colours using a picosecond laser." However, for the color palette In Fig. 1, only very limited color patterns are realized here. Especially the green and red colors are not represented well yet. Even for other colors, the color saturation and the color brightness are not good compared to other published papers in color printing using silver thin layers. The oxidization for silver sitting in air is another serious issue, the authors need to address this problem and analyze how long their sample can be stable in air without degrading the color performance. Also they need to give a solution to protect the sample. Furthermore, the reflection spectra measured from all the samples should be added in Fig. 4. From the simulated data in Fig. 4, it shows the reflection spectra only cover the wavelength from around 450nm to 700nm, which explains that not all the visible colors are realized. In addition, the authors fail to cite other color printing related literature correctly in their manuscript. More references should be added in. To conclude, I do not recommend the publication of the current manuscript, it might be more fitful for optics related journals, such as Optics Express.

Reviewer #2 (Remarks to the Author):

This work presents interesting results pertaining to the creation of reflective colored surfaces on noble metals such as silver, using raster scanned picosecond laser pulses. The use of picosecond pulses results in the formation of metallic nanoparticles on the metal surfaces. The authors show that the color of the surfaces is dominantly a function of the total fluence illuminating the surface, which leads to a straightforward method for color control.

Several points make this work interesting. Firstly, the surface color is angle independent, which is usually not the case for other types of surface texturing approaches (e.g., in dielectrics). In addition, the treated surface areas are large in area and need not be planar. Furthermore the authors have carried out extensive material characterization, statistical analysis and numerical simulations in order to understand and analyze the fabrication process and the physics behind the color of the samples.

The fact that the color is also dependent on effects such as surface passivation does make the structures good candidates for biosensing applications. The results should be of interest to the optical science community, in particular researchers working in plasmonics, laser fabrication methods and optical sensing. The general public may also find this work to be of interest, especially those working in the jewelry area.

The actual laser manufacturing process seem to be quite straightforward and reproducible by a third party researcher.

The paper is presenting solid experimental results backed by convincing and in-depth simulations and analysis and I believe that it is a good candidate for being published in Nature Communications.

Reviewer #3 (Remarks to the Author):

Topographical coloured plasmonic coins

By Jean-Michel Guay et al.

1. The paper contains interesting physics on plasmonics.
2. The quality of the research work presented in the paper is very good.
3. The paper is scientifically sound.
4. The paper is poorly written and it cannot be understood by nonspecialists.
5. The abstract is also poorly written. The abstract should include the method used and new results found clearly.
6. How the shape of nanoparticles will affect the intensity and frequency of the reflectance spectra?
7. How the results of this paper will be modified if one uses metallic nanoshells?
8. It is not clear whether authors are referring to surface plasmon polariton (SPP) resonances as plasmonic cluster resonances. It is well known the SPP resonances depend on the shape and size of nanoparticles. How the resonance frequencies and the reflectance spectra will be modified if one changes the shape of clusters.
9. It is not clear whether authors have included the dipole-dipole interaction between cluster or between nanoparticles in their simulations? What will be its effect on the reflectance spectra?
10. The finding of the paper depends on dielectric constant of the metallic nanoparticles and clusters. How this quantity has been calculated? Generally, the Drude model is used to evaluate the dielectric constant. However, this model is not valid for nanoparticles.
11. The thermal decay rate appearing in the dielectric constant plays a very important role in plasmonics. How the decay rate has been calculated?
12. In my opinion, the results of this paper do not quite meet the impact and innovation criteria of this journal. Impact of this work on the plasmonics field will be low.

Finally, I recommend that the paper should not be accepted for the publication in the present form.

Dear Reviewers,

We thank you for taking the time to carefully read our manuscript and for the valuable comments you have provided, which helped us in improving the revised paper that we are re-submitting for review.

Please find below our detailed response to each of the comments.

Best regards,

Jean-Michel Guay

Reviewer 1:

Recommendation: Not suitable for publication

Comments:

“The authors have used ps laser to make nanoparticles on silver surface so that different visible colors can be realized due to the light absorption of nanoparticles with different sizes on the surface. First, as the authors agreed the original idea of making such color pattern on metal surface is not new, as the references cited in the manuscript, Refs. 18-20. Instead, the authors just try to tune the set of laser parameters of the laser machining to get more colors. The whole manuscript is written in a way to be too technical but not scientific enough. It discusses a lot of technical part, such as what is the ps laser parameters and lots of experimental characterization details. Also although the authors claimed that their method is “producing a complete angle-independent colour palette composed of thousands of colours using a picosecond laser.” However, for the color palette In Fig. 1, only very limited color patterns are realized here. Especially the green and red colors are not represented well yet. Even for other colors, the color saturation and the color brightness are not good compared to other published papers in color printing using silver thin layers. The oxidization for silver sitting in air is another serious issue, the authors need to address this problem and analyze how long their sample can be stable in air without degrading the color performance. Also they need to give a solution to protect the sample. Furthermore, the reflection spectra measured from all the samples should be added in Fig. 4. From the simulated data in Fig. 4, it shows the reflection spectra only cover the wavelength from around 450nm to 700nm, which explains that not all the visible colors are realized. In addition, the authors fail to cite other color printing related literature correctly in their manuscript. More references should be added in. To conclude, I do not recommend the publication of the current manuscript, it might be more fitful for optics related journals, such as Optics Express.”

Response to each comment of Reviewer 1 (individually):

First, as the authors agreed the original idea of making such color pattern on metal surface is not new, as the references cited in the manuscript, Refs. 18-20. Instead, the authors just try to tune the set of laser parameters of the laser machining to get more colors.

[We agree that the pursuit of a direct laser coloring process on metals has been of some interest in the literature.]

We are aware of only one group that has worked on this topic with picosecond lasers, Refs. 20, 21 (Refs. 19, 21 in our original manuscript), indeed reporting very nice work on colours created on the surface of metal, in that case copper. However, the colours produced were rather limited, and of very low Chroma. Plasmonic effects were assumed to be responsible but there was no theoretical investigation to support this claim. Also, other work on copper (from a different group) showed that oxide produced on copper could generate the same colours (Ref. 22).

The other cited references, Refs. 18, 19 (Refs. 18, 20 in our original manuscript) refer to work done with femtosecond lasers. The processes reported have a low throughput, and thus are of limited interest for industrial applications. Furthermore, the majority of the colours produced are angle-dependent due to underlying periodic structures that are also created in the process. It is unclear whether the few angle-independent colours produced are plasmonic in nature, as it they could have been created via oxides or aggregates (see – Jwad, T. *et al.*, “Laser induced single spot oxidation of titanium,” *Applied Surface Science*, **387**, 2016).

We believe that the first version of our manuscript had already presented several original ideas, including a colour palette containing every hue, the demonstration of a link between Hue and total accumulated fluence (*i.e.*, the existence of a master curve), and a computational model proving that plasmonic excitations are key to colour production. None of this has been approached in any prior publication. The existence of a master curve, for example, has a great impact on the industrial applicability of laser colouring (as we discuss in our paper) because the throughput can be significantly increased via a trade-off against laser fluence.

However, we have made major additions to our manuscript, adding more original content, further distinguishing our paper from previous work, including the following:

1. We have expanded our laser processing technique to include multiple short-time-delay bursts of ps laser irradiation which has not been considered elsewhere in this context and which we find greatly extends the range of colours that can be produced. We now report colours of significantly increased Chroma (saturation), and of significantly increased range of Lightness, using burst irradiation.
2. In addition to presenting new and significantly expanded colour palettes on silver, we have added colouring results on copper, aluminum and gold. Like silver, burst irradiation is observed to significantly increase the saturation of the colours on these metals. Our palettes of (viewing) angle-independent colours on gold and copper are a bit surprising given the existence of an absorption edge in these metals in the visible. Our green, purple and blue gold are particularly interesting, and will be investigated theoretically in future work. Our colouring results on these metals (copper, gold, aluminum) suggest the existence of a master curve (as for silver), which will also be pursued in future work.
3. We have added colouring results on a 5 kg silver collectible coin, 21 cm in diameter, 2.5 cm thick, having topographic features spanning 1.5 cm (peaks and valleys) as an additional demonstration. To our knowledge, no other technique (bottom-up or top-down) is capable of colouring such a large and complex surface.
4. We have obtained reflectance measurements from coloured silver surfaces, which we compare to reflectance curves calculated from simulations. We find our theoretical model based on the statistical

distributions of nanoparticles is well able to reproduce experimental trends and explain the origin of the colours.

We hope that these clarifications and major additions will now convince the reviewer of the novelty of our work.]

The whole manuscript is written in a way to be too technical but not scientific enough. It discusses a lot of technical part, such as what is the ps laser parameters and lots of experimental characterization details.

[We respectfully disagree with the reviewer on this comment, as we feel there was significant scientific content in the original manuscript. For example, we carry out extensive SEM imaging and statistical analyses to understand the nature of the nano-structuring produced by the irradiation process. The theoretical section of the paper then reports the extensive electromagnetic modelling of model surfaces informed by the SEM images and statistics, demonstrating the importance of the bimodal distribution of the nanoparticles on the creation of colours, making also the connection with plasmonic excitations thereon, and highlighting the essential role of plasmons in rendering colour.

Nevertheless, we have made some changes in response to this comment. We have reorganized the manuscript, where the application and technical description of the process are now reported in Sub-section 3.1 and in the methods section, respectively. The remaining sections of the paper (*i.e.*, most of the body) are now primarily scientific in nature, dedicated to describing, interpreting and explaining our colouring results.

Also, many of the additions during revision increase further the scientific content, particularly in the expanded theoretical section, where the origin of the range of colours is determined in terms of nanoparticle distributions, and agrees very well with observed experimental trends.]

Also although the authors claimed that their method is “producing a complete angle-independent colour palette composed of thousands of colours using a picosecond laser.” However, for the color palette In Fig. 1, only very limited color patterns are realized here. Especially the green and red colors are not represented well yet.

[To address this comment and better support our claims, we have added several additional colour palettes which now include red and green colours. As a result, Figs. 2 and 3 have been significantly modified and expanded. Fig. 2(d) in particular shows a more detailed and complete palette obtained using our colouring processes, with the colour bar along the top of the figure constructed from photographs of the colours obtained. Thousands of colours with different LCH values have been gathered to produce Figures 2 and 3. The green and red colours are now well-represented.]

Even for other colors, the color saturation and the color brightness are not good compared to other published papers in color printing using silver thin layers.

[To address this comment, we have added more colouring results, as summarised in new Fig. 3. In particular, the burst laser irradiation process added to the paper produces colours that have significantly higher Chroma (Fig. 3(a)) and a significantly larger Lightness range (Fig. 3(b)), resulting in expanded coverage of colours as shown on the new chromaticity diagram of Fig. 3(c). Our results compare very favourably to the best results achieved to date using nanoscale bottom-up methods. We have added points

of comparison in terms of the results achieved, and points of differentiation in the techniques applied, at the end of sub-section 3.2.]

The oxidization for silver sitting in air is another serious issue, the authors need to address this problem and analyze how long their sample can be stable in air without degrading the color performance. Also they need to give a solution to protect the sample.

[We had addressed this topic in the original manuscript, but have expanded the discussion in the revised paper (see last paragraph of sub-section 3.1). We protected our coloured coins through the deposition of a passivation layer formed using ALD (atomic layer deposition). ALD-passivated coloured coins were then subjected to aggressive tarnish, humidity and temperature tests (standardised tests applied at the Royal Canadian Mint), showing no evidence of degradation whatsoever. We are presently working on a separate manuscript describing our ALD passivation process and the environmental tests carried out.]

Furthermore, the reflection spectra measured from all the samples should be added in Fig. 4.

[We have added some measured reflectance spectra to the manuscript in new Fig. 8.]

From the simulated data in Fig. 4, it shows the reflection spectra only cover the wavelength from around 450nm to 700nm, which explains that not all the visible colors are realized.

[The reviewer is referring to old Fig. 4(b) which is now relabelled as Fig. 6(b). This figure was produced for only one distribution of nanoparticles corresponding to a single colour (*i.e.* for a single line spacing), where the inter-particle distance is fixed and only the embedding of the small nanoparticles is varied in depth by increments of 0.5 nm. The purpose of this figure is solely to show the sensitivity of the system to embedding of the small nanoparticles.

The full spectrum of colours perceived originates from varying the nanoparticle distribution on the surface by varying the laser write parameters. In new Fig 8(b), we now present calculated reflectance spectra that span the entire visible range, for several nanoparticle distributions, each leading to a different colour.]

In addition, the authors fail to cite other color printing related literature correctly in their manuscript. More references should be added in.

[Our original list of references acknowledged the work of many groups working on plasmonic colours and on the laser processing of metals. We also believe our summary of the previous literature to be correct. Nevertheless, we have added a few more references (Refs. 35, 36, 37) including a review article that was published while revising our paper (new Ref. 7).]

Reviewer 2:

Recommendation: Suitable for publication in its current form

Comments:

“This work presents interesting results pertaining to the creation of reflective colored surfaces on noble metals such as silver, using raster scanned picosecond laser pulses. The use of picosecond pulses results in the formation of metallic nanoparticles on the metal surfaces. The authors show that the color of the surfaces is dominantly a function of the total fluence illuminating the surface, which leads to a straightforward method for color control.

Several points make this work interesting. Firstly, the surface color is angle independent, which is usually not the case for other types of surface texturing approaches (e.g. , in dielectrics). In addition, the treated surface areas are large in area and need not be planar. Furthermore the authors have carried out extensive material characterization, statistical analysis and numerical simulations in order to understand and analyze the fabrication process and the physics behind the color of the samples.

The fact that the color is also dependent on effects such as surface passivation does make the structures good candidates for biosensing applications. The results should be of interest to the optical science community, in particular researchers working in plasmonics, laser fabrication methods and optical sensing. The general public may also find this work to be of interest, especially those working in the jewelry area.

The actual laser manufacturing process seem to be quite straightforward and reproducible by a third party researcher.

The paper is presenting solid experimental results backed by convincing and in-depth simulations and analysis and I believe that it is a good candidate for being published in Nature Communications.”

[We thank the reviewer for studying our paper and providing positive feedback and comments.]

Reviewer 3:

Recommendation: Not suitable for publication

Comments and our response:

“Topographical coloured plasmonic coins

By Jean-Michel Guay et al.

1. The paper contains interesting physics on plasmonics.
2. The quality of the research work presented in the paper is very good.
3. The paper is scientifically sound.
4. The paper is poorly written and it cannot be understood by nonspecialists.

[We do not agree that our manuscript was poorly written. However, we have re-organised the material and expanded our explanations in revising our paper, which should make the subject more accessible to non-specialists.]

5. The abstract is also poorly written. The abstract should include the method used and new results found clearly.

[Our revised abstract is now more concise, to the point, and includes the elements mentioned by the reviewer.]

6. How the shape of nanoparticles will affect the intensity and frequency of the reflectance spectra?

[This is an interesting question which could be the subject of further study. In our model, we constructed surfaces comprising small and medium nanospheres, and portions of nanospheres created by embedding, inspired by SEM images of surfaces (*e.g.*, Fig. 5(j)) and their statistics. This model is reasonable and our theoretical results support the experimental data and our interpretation.]

7. How the results of this paper will be modified if one uses metallic nanoshells?

[Using metallic nanoshells would certainly alter the results but we see no reason to consider them on the coloured surfaces of interest in this paper.]

8. It is not clear whether authors are referring to surface plasmon polariton (SPP) resonances as plasmonic cluster resonances. It is well known the SPP resonances depend on the shape and size of nanoparticles. How the resonance frequencies and the reflectance spectra will be modified if one changes the shape of clusters.

[We clarify in the revised manuscript that we are referring to collective resonances on arrangements of nanoparticles. We have expanded our discussion of the effects mentioned by the reviewer and added a

new figure to our theoretical section summarising the effects of several shape variations on the reflectance (Fig. 8(b)).]

9. It is not clear whether authors have included the dipole-dipole interaction between cluster or between nanoparticles in their simulations? What will be its affect on the reflectance spectra?

[The close proximity of the nanoparticles in our arrangements leads to strong electromagnetic coupling among them and to collective resonances. These electrodynamic effects are modeled rigorously in our implementation of the FDTD method (full wave, three dimensions).]

10. The finding of the paper depends on dielectric constant of the metallic nanoparticles and clusters. How this quantity has been calculated? Generally, the Drude model is used to evaluate the dielectric constant. However, this model is not valid for nanoparticles.

[We model dispersion in silver via the Drude model enriched by 2 critical points (Drude+2CP model) which can accurately model the dispersion even in the interband region of silver. The validity of a macroscopic model begins to break for small nanoparticles (diameter < 5 nm and gap sizes ~ 1 nm) due to the presence of nonlocal effects. The size of our smallest nanoparticles (diameter ~ 7 nm) and smallest edge-to-edge particle inter-distance (~ 3 nm) are thus still within the limit of validity of the Drude+2CP model.]

11. The thermal decay rate appearing in the dielectric constant plays a very important role in plasmonics. How the decay rate has been calculated?

[We used measured data for the dielectric constant of silver to which the Drude+2CP model was fitted, thereby extracting equivalent parameters including the decay (electron scattering) rate.]

12. In my opinion, the results of this paper do not quite meet the impact and innovation criteria of this journal. Impact of this work on the plasmonics field will be low.

[We refer the reviewer to our response to the first comment of Reviewer 1. We believe that our revised manuscript now meets the impact and innovation criteria of Nature Communications.]

Finally, I recommend that the paper should not be accepted for the publication in the present form.

Reviewers' Comments:

Reviewer #1:

Remarks to the Author:

The authors have shown a lot of efforts to improve the manuscript and this should be well appreciated. I found the authors have addressed all my comments carefully and in detail by adding more materials in the text and more figures. As a result, I now recommend the current form can be accepted for publication without further modification.

Reviewer #2:

None

Reviewer #3:

Remarks to the Author:

Topographical coloured plasmonic coins

By Jean-Michel Guay et al.

I went through the revised paper and referees report along with authors reply very carefully. I have the following comments.

1. The revised version has been improved significantly and authors have done an excellent job to revised the paper.
2. However, authors have not answered my questioned and comments satisfactorily.
3. The revised paper is still very technical and cannot be understood by non-specialists.
4. I agree with referee 1 that the paper might be more suitable for publication in optics related journals, such as Optics Express.

Finally, I recommend that the paper should NOT be accepted for the publication since it does not meet the criterion such as impact, innovation and interest for this journal.

Dear Editor and Reviewers,

We would like to thank you for taking the time to carefully read our manuscript and for the feedback provided. Please find below our response to each comment.

Best regards,

Jean-Michel Guay (corresponding author)

Reviewer 1:

Recommendation: Suitable for publication in its current form

Comments:

“The authors have shown a lot of efforts to improve the manuscript and this should be well appreciated. I found the authors have addressed all my comments carefully and in detail by adding more materials in the text and more figures. As a result, I now recommend the current form can be accepted for publication without further modification.”

[We thank the reviewer for studying our revised manuscript and for providing positive feedback and comments. We thank Reviewer 1 for providing comments on the previous version which helped us strengthen the manuscript to its current form.]

Reviewer 3:

Recommendation: Not suitable for publication

Comments:

“I went through the revised paper and referees report along with authors reply very carefully. I have the following comments.

1. The revised version has been improved significantly and authors have done an excellent job to revised the paper.
2. However, authors have not answered my questioned and comments satisfactorily.
3. The revised paper is still very technical and can not be understood by non-specialists.
4. I agree with referee 1 that the paper might be more suitable for publication in optics related journals, such as Optics Express.

Finally, I recommend that the paper should NOT be accepted for the publication since it does not meet the criterion such as impact, innovation and interest for this journal.”

Response to each comment of Reviewer 3 (individually):

“1. The revised version has been improved significantly and authors have done an excellent job to revised the paper.”

[We thank Reviewer 3 for acknowledging that our manuscript is significantly improved and we thank him for commenting on the previous version which helped us strengthen the manuscript to its current form.]

“2. However, authors have not answered my questioned and comments satisfactorily.”

[We disagree and believe that we addressed all of the comments of Reviewer 3 satisfactorily, either through rebuttal or edits to the manuscript, without deviating from the core of our manuscript.]

“3. The revised paper is still very technical and can not be understood by non-specialists.”

[We believe that our revised manuscript is now organised and written in such a way as to attract a broad readership. Furthermore, we modified (lightened) the theoretical section of the manuscript in order to make the paper less technical, without altering the interpretations and conclusions.]

“4. I agree with referee 1 that the paper might be more suitable for publication in optics related journals, such as Optics Express.”

[We point out that Reviewer 1 has now recommended publication of our revised manuscript in its current form in Nature Communications.]